# New ring shear deformation apparatus for three-dimensional multiphase experiments: First results

Shae McLafferty[1], Haley Bix[1], Kyle Bogatz[1], Jacqueline E. Reber[1]

[1]Department of Geological and Atmospheric Sciences, Iowa State University, Ames, Iowa, 50011, USA

*Correspondence to*: Jacqueline Reber (jreber@iastate.edu)

Key words: Ring shear apparatus, HydroOrbs, Hydrogel spheres, Semi-brittle experiments, granular experiments

**Abstract**

Multiphase deformation, where a solid and fluid phase deform simultaneously, plays a crucial role in a variety of geological hazards, such as landslides, glacial slip, and the transition from earthquakes to slow slip. In all these examples a continuous,

viscous or fluid-like phase is mixed with a granular or brittle phase where both phases deform simultaneously when stressed. Understanding the interaction between the phases and how they will impact deformation dynamics is crucial to improve the hazard assessments for a wide variety of geo-hazards. Here, we present the design and first experimental results from a ring shear deformation apparatus capable of deforming multiple phases simultaneously. The experimental design allows for three dimensional observations during deformation in addition to unlimited shear strain, controllable normal force, and a variety of

boundary conditions. To impose shear deformation, either the experimental chamber or lid rotate around its central axis while the other remains stationary. Normal and pulling force data are collected with force gauges located on the lid of the apparatus and between the pulling motor and the experimental chamber. Experimental materials are chosen to match the light refraction index of the experimental chamber, such that 3D observations can be made throughout the experiment with the help of a laser light sheet. We present experimental results where we deform hydropolymer orbs (brittle phase) and Carbopol® hydropolymer

gel (fluid phase). Preliminary results show variability in force measurements and deformation styles between solid and fluid end member experiments. The ratio of solids to fluids and their relative competencies in multiphase experiments control deformation dynamics, which range from stick-slip to creep. The presented experimental strategy has the potential to shed light on multi-phase processes associated with multiple geo-hazards.

**Plane language abstract (500 characters incl. space)**

Multiple geologic hazards, such as landslides and earthquakes, arise when solids and fluids coexist and deform together. We designed an experimental apparatus that allows us to observe such deformation in 3D. The first results show how fluids and solids deform and break at the same time allowing us to study the impact of both materials on deformation distribution and speed. Making these processes visible has the potential to improve risk assessments associated with geological hazards.

# 1 Introduction

Multiple geo-hazards result from three key ingredients: A solid phase that can fracture unstably, a fluid phase that influences the state of stress and can have a viscosity spanning many orders of magnitude, and a driving force such as gravity or tectonics. For example, on hillslopes, incipient shear of sediments creates volume change, which in turn causes the pore-water flow and the associated stress changes that govern the stability of landslides (e. g., Iverson et al., 2000). At crustal scales, the proportions of solid to fluid phases and their interactions can modulate deformation dynamics and lead to a spectrum of behaviour from earthquakes to slow slip events (e. g., Fagereng and Sibson, 2010; Behr and Bürgmann, 2021; Kirkpatrick et al., 2021). In addition, syn-deformation solid-fluid interactions control slip rates at the beds of ice sheets (e. g., Iverson et al., 1995; Zoet and Iverson, 2020) — the single most significant uncertainty in predicting dynamic contributions of ice sheets to sea-level rise over the next century as the climate warms (Stocker and Others, 2013; Rignot et al., 2019). In all these examples a continuous, viscous or fluid-like phase is mixed with a granular or brittle phase where both phases deform simultaneously when stressed. Understanding the interaction between the phases and how they will impact deformation dynamics is crucial to improve the hazard assessments for a wide variety of geo-hazards.

While there are many experimental (e.g., Ladd and Reber, 2020; Reber et al., 2014; Higashi and Sumita, 2009) and numerical studies (Ioannidi et al., 2022; Jammes et al., 2015; Ioannidi et al., 2021; Behr et al., 2021) that investigate different aspects of two-phase or brittle-viscous interactions, they face multiple challenges and limitations. To resolve the complex interaction of the brittle and viscous phase, high resolution experiments or simulations are necessary. In addition, the materials need to be able to deform in different manners independently of each other. This means that the brittle material loses cohesion when failure occurs while the viscous material flows under stress. Furthermore, the impact of simultaneous two-phase deformation is inherently a three-dimensional problem. Numerical experiments are suitable to evaluate two-phase systems in 3D where it is possible to make continuous observations. In addition, systematic parameter studies are feasible. However, the resolution of numerical models, especially in 3D is strongly dependent on available computational resources. But perhaps the greatest drawback of numerical models is the difficulty of having two phases where one is continuous and the other is able to break and therefore becoming discontinuous. The simultaneous deformation of two fundamentally different phases is trivial in physical experiments, as is the resolution issue. While the scaling of experiments using analogue materials remains a challenge, a further hurdle is the observation in 3D. It is difficult to make observations in 3D without the need to destroy the experiment by slicing it open and therefore limiting the deformation progression.

Here, we present the design of a new ring shear deformation apparatus that allows deformation of multiphase experiments to be monitored in 3D. Besides the apparatus design, data acquisition process, and visualization of the 3rd dimension in experiments, we present the first data gathered with the device to demonstrate its versatility and potential applications.

## 2 Shear apparatus

### 2.1 Apparatus design

The ring shear apparatus is designed for the purpose of observing and quantifying deformation of multiphase materials. While the shear apparatus may resemble a large rheometer, it however, serves a different purpose. It is designed to deform mixtures of experimental materials with known physical properties in three dimensions, resolve internal deformation, and track the force driving deformation with a force gauge. This allows us to compare the impact of the experimental material on deformation dynamics and distribution.

The apparatus has no theoretical limits on applied strain, has controllable normal force (confining pressure), and is combined with an optical setup to make observations of internal deformation while the experiment is in progress. The apparatus consists of an experimental chamber, a hydraulically controlled lid that exerts a normal force, and a motor that initiates shear by rotating the experimental chamber (Table 1). The experimental chamber is built with two concentric transparent cylinders to form a ring-shaped gap (an annulus in a two-dimensional plan view). The radii of the cylinders are 19 cm and 11 cm for the outer and inner cylinders, respectively, resulting in an 8 cm wide annulus. Both cylinders have a height of 16 cm, of which approximately 14 cm can be filled with the experimental materials. The cylinders are sealed to a baseplate at the bottom thus both the cylinders and baseplate move as a unit during deformation. A hydraulically controlled lid can be lowered between the walls of the experimental chamber and onto the experimental materials to exert a normal force (Figure 1). The hydraulic system controls the lid to either exert a constant pressure or hold the lid at a constant position. The gaps between experimental chamber and the lid are sealed with o-rings that are lubricated with grease to reduce friction between the chamber walls and the lid. Normal force is recorded with a force gauge connected to the top of the lid.

An electromagnetic rotary motor encompasses the baseplate of the experimental chamber (grey circle in Figure 1) and rotates at a steady angular velocity. The motor is connected to the experimental chamber (Figure 1) to transfer motion. The difference in motion between the lid and the experimental chamber results in shear of the materials. The apparatus is designed so that the experimental chamber can be turned while the lid remains stationary, leading to a Eulerian observational system in which the observation window is stationary while the experimental material passes through it. This allows for the observation of spatial variability. However, the apparatus can be configured in such a way that the experimental chamber is stationary and the lid turns. This allows for Lagrangian observations in which the evolution of one parcel of the material can be observed during increasing shear deformation. Eight teeth, 1 cm high and wide, transverse both the lid and baseplate and help to transfer motion onto the experimental material. The most relevant specification and sources of third-party components used to build this machine are listed in Table 1.

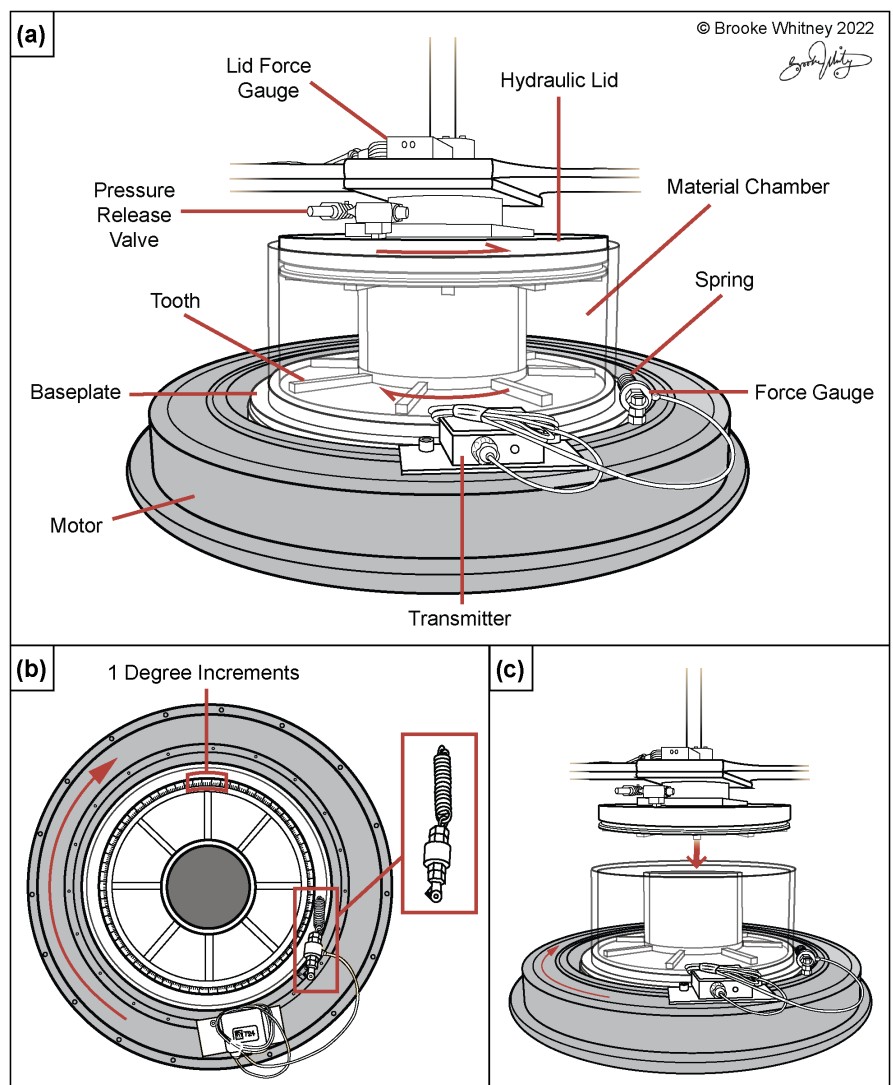

**Figure 1: Illustration of the ring shear apparatus named *Shearknado*. A) Labelled side view of the apparatus with the lid lowered into the experimental chamber. B) Plan view of the experimental chamber and surrounding motor with a close-up of spring and force gauge configuration. C) Side view of the apparatus. Illustration credit to ©Brooke Whitney 2022.**



**Table 1: Source and most relevant specifications of parts provided by third party sellers for the shear apparatus.**

|  | Source: | Specifications: |
|---|---|---|
| Direct drive rotary servo table | Intelidrives | max. velocity: 108 rpm |
| Air-powered high-cycle, high-flow hydraulic system | Milwaukee Cylinder | Pressure: 11000-16000 kPa |
| Amplified load cell (normal force) | Interface | Sensor capacity: 69000 kPa |
| Miniature in-line load cell (pulling force) | Applied measurements | max. load: 100 N |
| Diode pumped green laser | CrystaLaser | Wavelength: 532 nm |
|  |  | Output power: 100 mW |

## 2.2 Boundary condition

The ring shear apparatus allows for either a constant strain rate or an energy conserving boundary condition. For the constant strain rate boundary condition, the experimental chamber is connected directly to the motor with a force gauge in between. The force gauge records the bulk force required to rotate the material chamber, and the strain rate is set by the rotation velocity of the motor. Conversely, the energy conserving boundary condition neither prescribes the strain rate nor the stress (Birren and Reber, 2019; Daniels and Hayman, 2009; Reber et al., 2014). In this case, the experimental chamber is connected to the

force gauge and the motor via a spring (Figure 1b). Adding the spring creates a boundary condition that allows for strain rate and force to vary as the spring extends or contracts in response to material deformation (Figure 2 a). Continuous deformation, or creep, in the material chamber results in a relatively smooth force signal. The spring first elastically loads, increasing the pulling force until the spring is fully loaded, followed by minimal oscillation of the spring. Conversely, frictional deformation or stick-slip in the material chamber results in noticeable spring oscillation after initial loading. Shear within the experimental

chamber only occurs once the frictional resistance of the apparatus and the experimental material strength are overcome. This leads to repeated increases of force followed by decreases resulting in stick-slip like motion (Figure 2 a).

Force gauge measurements are taken at a frequency of 10 Hz. The recorded force signal is dependent on the stiffness of the spring (Figure 2 b and c) that has to be chosen according to the weight of the experimental material. At a minimum, the spring needs to be strong enough to be able to pull the loaded experimental chamber. A large spring constant will lead to smaller and

sharper peaks in the force curve where, in an extreme case, a constant strain rate boundary condition is approached. A small spring constant leads to a noisier signal. For the experiments presented here, we chose a spring with a constant of 9712 N/m. This spring constant and force measurement frequency combination ensures we can fully capture any force signal resulting from material deformation from creep to stick-slip, including transient deformation.

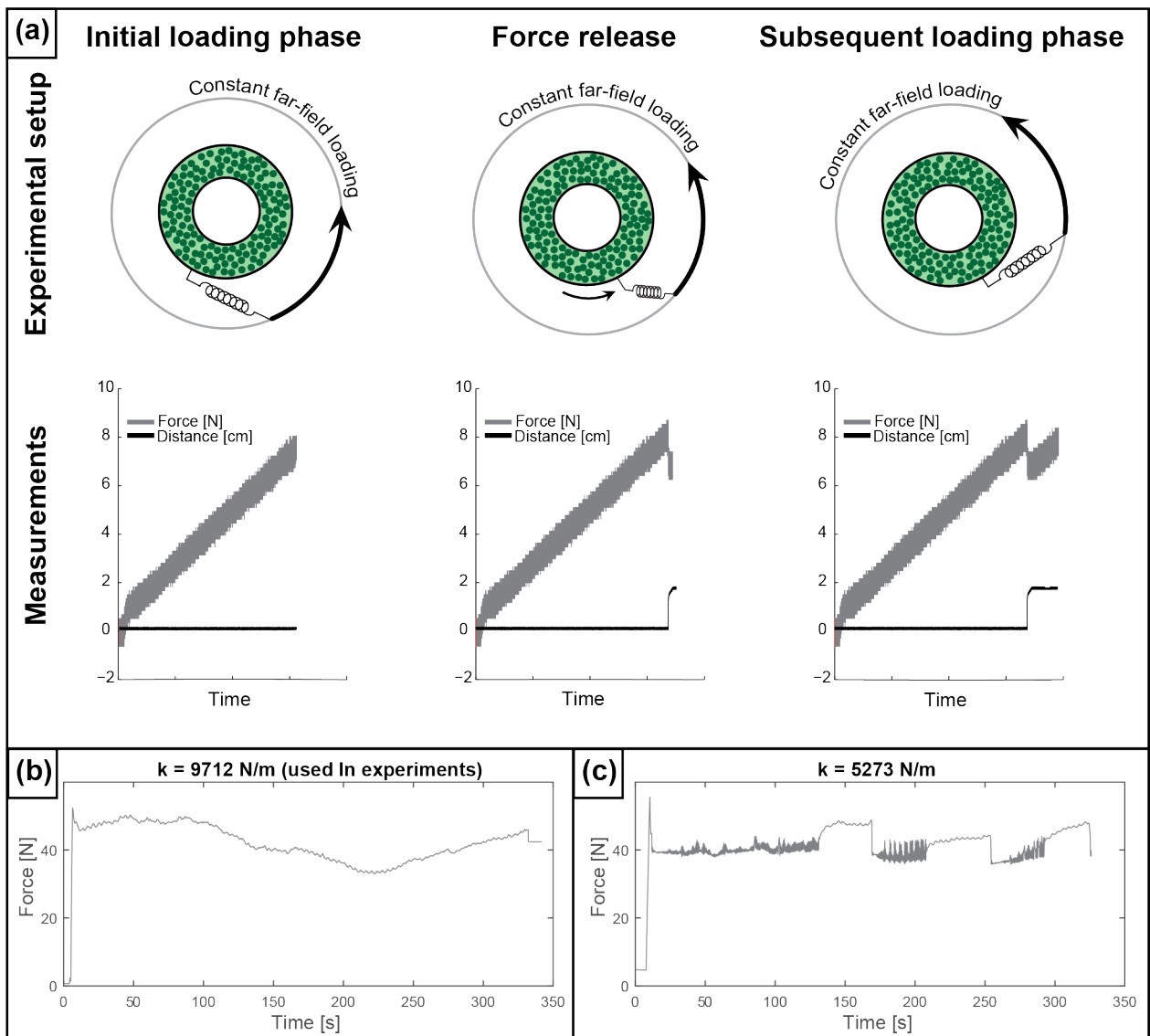

**Figure 2: a) Plan-view schematic of the ring shear apparatus illustrating the energy conserving boundary condition. (Left) Initial loading accumulates force until slip occurs. (Middle) Slip-event results in force drop and sharp increase in displacement. (Right) Subsequent loading accompanied by an increase in force and no motion. b) Force measurement from semi-brittle experiment using a spring with spring constant k = 9712 N/m. c) Force measurement from same experiment with spring constant k = 5273 N/m.**

## 2.3 Observation of internal deformation

The experimental chamber walls are made of transparent acrylic plastic allowing for 360-degree observation of the experiment (Figure 3). To document the deformation within the chamber we take advantage of the almost identical light refraction indices

of the experimental materials and the experimental chamber (Budwig, 1994; Byron and Variano, 2013; Klein et al., 2013;

Dijksman et al., 2017). Material mixtures with identical or very similar light refraction indices allow light to travel through the entire experimental chamber. If the refraction indices of the individual phases are different, one phase will cast a shadow and/or scatter the light and obscure the other phase. Carbopol, HydroOrbs, and transparent acrylic plastic are all transparent with similar refractive indices of 1.33-1.35, 1.333 (very close to water), and 1.490, respectively (Auernhammer et al., 2020; Parker

and Merati, 1996). This makes any mixture of these materials indistinguishable in natural light. Illumination with a laser light sheet, however, makes the different phases visible due to small differences in the light refraction indices (Figure 4). While the difference in light refraction indices between materials is large enough to make the different phases visible in laser light, it is small enough to not cast any shadows, resulting in the illumination of an entire slice through the experimental chamber (Mukhopadhyay and Peixinho, 2011).

The laser sheet originates from a 100 mW class 3B, 532 nm laser located 1 m from the experimental chamber (Figure 3 a). These specifications allow for the beam to illuminate an entire cross section of the experiment without excessive heating. To create the laser sheet, a series of optical lenses and mirrors are used to manipulate the beam (Figure 3 c and d). Alignment of the beam is controlled by using two 20 mm round silver mirrors placed at 45-degree angles from the beam path. Two spherical lenses with focal lengths of 50 mm and 200 mm are placed 150 mm apart within the final beam trajectory to magnify (4x) and

collimate the beam. A -25 mm focal length convex cylindrical lens then expands the beam, forming a vertical sheet. A 1000 mm focal length spherical lens is the final optic placed in the beam path, 1m away from the experimental chamber, to create a narrow beam at the point of penetration into the experiment chamber. During experimentation, a camera is placed perpendicular to the laser sheet and captures cross sectional photos as the chamber rotates.

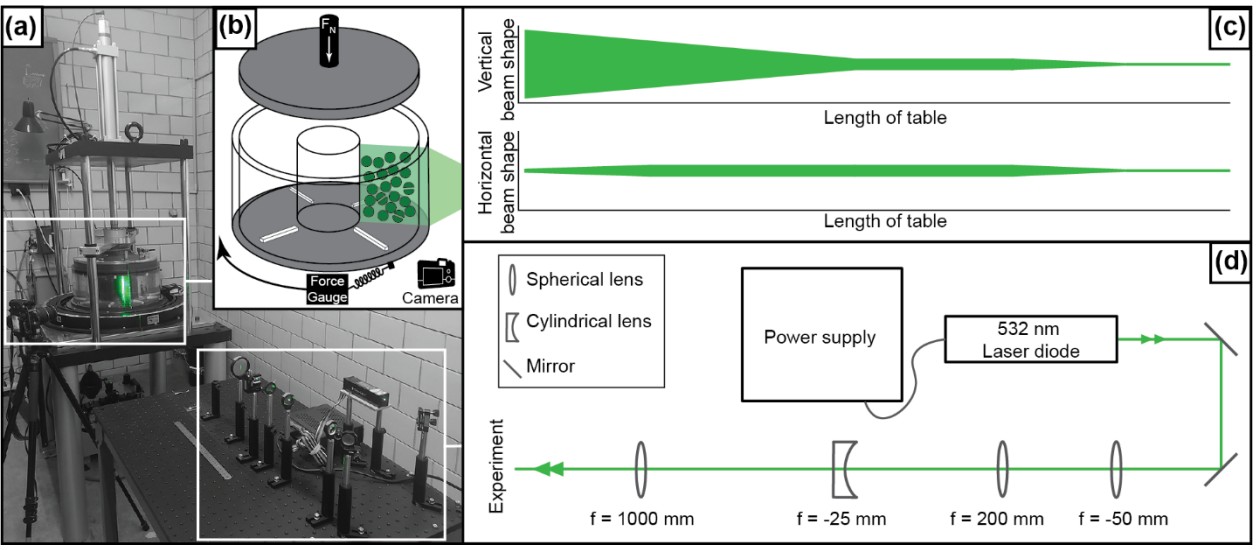


**Figure 3: a) Photograph of experimental setup, b) closeup sketch of the experimental chamber with the laser sheet and camera position. c) Vertical and horizontal beam shape. d) Plan view showing the optical layout used to create the laser sheet. Over the length of the table, the first two lenses magnify and collimate the beam in both the vertical and horizontal planes. In the vertical direction, the third lense stretches the beam to create a laser sheet. The fourth lens thins the laser sheet in the horizontal plane.**

## 3 Experimental materials

The experimental materials presented here are not an exhaustive list of all potential materials that can be used in the apparatus for experiments, but rather a selection we have chosen to use. With experimental chamber walls made of transparent acrylic plastic all experimental materials must have similar optical properties to be visible within the experiment. So far, we have conducted experiments using Carbopol, HydroOrbs, and a mixture of both. While water has also a comparable light refraction index, we did not conduct experiments using water as its very low viscosity makes it difficult to avoid leakage.

### 3.1 Brittle phase (HydroOrbs)

HydroOrbs, also known as polymer hydrogel spheres (e. g., James et al., 2020), are elasto-plastic solids that deform elastically. Once a yield stress is reached they break (Figure 4). HydroOrbs, which are spherical in shape, begin as dehydrated plastic pellets. When the dehydrated orbs are placed in water, they incorporate $H_2O$ into their structure and swell to about ten times their initial size. They reach their maximum volume after 1-4 days submerged in water. Once the orbs are fully hydrated, they are transparent. The volume of hydrated HydroOrbs is limited by the initial volume of the dehydrated plastic pellets, which come in small (2 mm) and large (4 to 5 mm) sizes and correspond to the small and large fully hydrated orbs listed in Table 2. The final size of the orbs can be manipulated to some degree via the water salinity, where lower salinity leads to larger diameters (Table 2). We conduct experiments on orbs that are soaked in either deionized (DI) water or tap water.

Water salinity also has an impact on the yield stress of the orbs. The yield stress of the orbs is measured before they are placed into the experimental chamber. The force at which the orbs fail and the area over which the force is applied is measured for a representative sample of orbs. The yield stresses are then averaged and listed in Table 2. Our measured orb yield stresses are in the range of measurements obtained under variable loading rates (James et al., 2020, 75 to 175 kPa). Beyond the influence of water, the yield stress of the orbs can be lowered by puncturing them with a needle. The puncture introduces a line of weakness by penetrating a rind of denser material in the outermost 1-2 mm of the orb (Chang et al., 2018). The elastic properties of the orbs are also measured prior to use in the experiments. Young's modulus is obtained by deforming a representative sample number of the orbs with 3 different known forces (recorded with a force gauge) and the resultant strain and area over which the force is applied is measured. Young's moduli values in the HydroOrb literature range from 10 to 100 kPa (e. g., Waitukaitis et al., 2017; Mukhopadhyay and Peixinho, 2011; Dijksman et al., 2017; Chang et al., 2018; James et al., 2020). Our measurements fall within this range (Table 2). The Poisson's ratio is calculated by deforming the orbs to a known height and the lateral and longitudinal strain is measured. Previous works have found the Poisson's ratios of the orbs to range from 0.3 to 0.45 (Chang et al., 2018), whereas others have idealized the Poisson's ratio to be 0.5 (James et al., 2020). The shear modulus is determined by the relationship between the Young's modulus and poison's ratio outlined in Gercek (2007).

An alternative material that can be used as the brittle phase in the ring shear apparatus are HydroCubes. Contrary to HydroOrbs, dehydrated HydroCubes are produced in large sheets, which can then be cut prior to hydration into desired size, shape, and

aspect ratio. By this premise, HydroCubes need not be cubes, but rather may be of any desired shape. They are limited only by the volume of the original HydroCube sheet. Future experiments will investigate the impact of HydroCubes as the brittle phase in the ring shear apparatus. For now, preliminary material properties of the cubes can be found in Table 2.


## 3.2 Viscous phase (Carbopol)

We use a visco-elasto-plastic hydropolymer gel, Carbopol® as the viscous phase in the experiments. Carbopol is a transparent, non-linear yield stress fluid with a power law viscosity that can be approximated the Herschel-Bulkley model in equation 1 (Herschel and Bulkley, 1926; Di Giuseppe et al., 2015),

$$\sigma = \sigma_y + K_v \dot{\varepsilon}^n \qquad (1)$$

where the stress, $\sigma$, is dependent on the yield stress, $\sigma_y$, the consistency index, $K_v$, the strain rate, $\dot{\varepsilon}$, and the flow index, $n$. The consistency index is a constant of proportionality between shear stress and strain rate where a higher consistency is a result of a greater change in shear stress from a change in strain rate (Reber et al., 2020). Both the yield stress and viscosity of the Carbopol can be adjusted by changing the polymer concentration and pH of the Carbopol gel mixture, respectively. We measure the shear stress and viscosity of the Carbopol gel used in experiments with a rheometer (Rheosys Merlin VR). The

range of Carbopol viscosities explored at the shear rate of the ring shear apparatus to date are listed in Table 2. This range of viscosities is achieved by manipulating the concentration of the polymer while keeping the pH of the gel consistent. The yield stress and viscosity values for the Carbopol gels measured fall in the range of values obtained in other studies (e. g., Di Giuseppe et al., 2015; Birren and Reber, 2019; Reber et al., 2015). For an extensive list of Carbopol gel properties, we direct the reader to Di Giuseppe et al., (2015). The Carbopol properties are held constant in the experiments presented here, with an

average yield stress, $\sigma_y$, of 28.21 Pa and average viscosity of ~240 Pa.s at experiment strain rates. The flow index, $n$, is calculated from the slope of the linear relationship between the logarithm of strain rate and the logarithm of shear stress and is found to be 0.37.



**Table 2: Experimental material properties. * Values from (Di Giuseppe et al., 2015).**

| Material Property | Small HydroOrb | | Large HydroOrb | HydroCube | Carbopol® |
|---|---|---|---|---|---|
| Water | DI | Tap | Tap | Tap | DI |
| | | | | | |
| Diameter (cm) | 1.68 ± 0.10 | 1.42 ± 0.09 | 3.87 ± 0.25 | - | - |
| Volume (cm^3) | 2.48 ± 0.45 | 1.54 ± 0.24 | 31.76 ± 8.27 | - | - |
| Mass (g) | 2.82 ± 0.43 | 1.73 ± 0.17 | 38.12 ± 11.27 | - | - |
| Density (g/cm^3) | 1.07 ± 0.05 | 1.08 ± 0.07 | 1.04 ± 0.06 | 1.05 ± 0.10 | 1.01 – 1.03* |
| Viscosity (Pa.s) | - | - | - | - | 101.76 - 448.91 |
| | | | | | |
| Poisson's Ratio | 0.39 ± 0.07 | 0.27 ± 0.08 | 0.37 ± 0.07 | 0.36 ± 0.10 | - |
| Young's Modulus (kPa) | 121.93 ± 57.59 | 143.72 ± 89.79 | 43.68 ± 23.16 | 19.72 ± 15.07 | - |
| Shear Modulus (kPa) | 42.98 ± 20.98 | 57.85 ± 38.36 | 16.78 ± 8.55 | 7.16 ± 5.58 | - |
| | | | | | |
| Yield Stress (kPa) | | | | | 0.01 - 0.05 |
| Non-punctured | 72.27 ± 18.86 | 78.55 ± 21.93 | 15.01 ± 14.37 | 5.92 ± 2.43 | - |
| Punctured | 25.62 ± 16.25 | 21.97 ± 15.70 | 7.07 ± 12.79 | - | - |


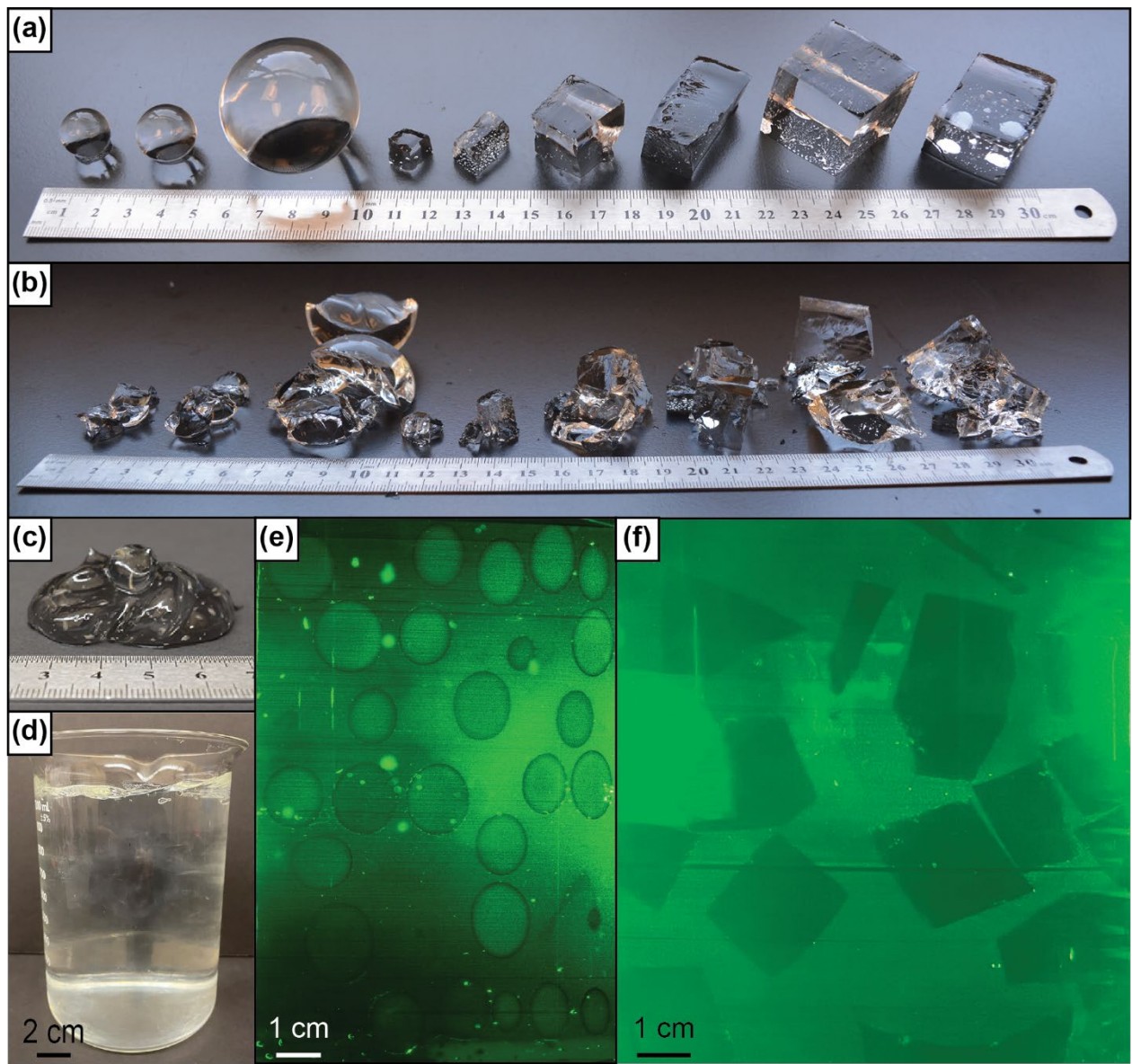

**Figure 4: a)** Fully hydrated HydroOrbs and HydroCubes. **b)** Fragmented HydroOrbs and HydroCubes after the yield stress is reached. **c)** Carbopol, **d)** Mixture of HydroCubes and Carbopol in natural light, **e)** HyrdoOrb and Carbopol mixture illuminated by the laser sheet. **f)** HydroCube and Carbopol mixture illuminated by the laser sheet. Note bright small spots are due to the light reflection on trapped air bubbles.


## 4 Data acquisition

### 4.1 Visual documentation

Internal deformation in semi-brittle experiments, where both phases are present in the experimental chamber, is recorded throughout the experiment with photos of the illuminated cross section. 360 photos of the illuminated cross-section are taken every one degree around the material chamber (Figure 1) before the experiment, after every rotation for the first 10 rotations, and then after rotations 15 and 20. Taking photos of the illuminated cross-section allows for the deformation in the brittle phase to be documented in the 3$^{rd}$ dimension. The 1-degree interval is small enough that every HydroOrb is captured in multiple

photos and ensures fragments of the broken orbs are also captured. After each experiment, broken orbs are counted and their locations within the illuminated cross section are recorded. This method ensures deformation in the brittle phase is recorded around the entire experimental chamber, in the third dimension, and with increasing strain over the duration of the experiment. We can then quantify deformation in the brittle phase and quantitatively compare results across experiments.

Cross-sectional photos only work for experiments where the entire experimental chamber is filled with material of similar light

refraction indices. For all other experiments (brittle and viscous experiments), pictures are taken perpendicular to the experimental chamber wall.

### 4.2 Force measurements

Normal force measurements are recorded with the force gauge located on the hydraulically driven lid at a frequency of 100 Hz. To measure the pulling force, a force gauge is mounted in series with a spring between the experimental chamber and the

rotating motor. The force gauge is connected to a wireless transmitter allowing for data collection over many rotations. The recorded pulling force is a bulk measurement consisting of the force required to deform the experimental material plus the frictional resistance of the experimental apparatus.

A background experiment is used to identify and separate the noise originating from machine friction from the signal of the deforming experimental material. The background experiment is performed by loading the experimental chamber with weights

comparable to the weight of the experimental materials intended to deform during an experiment. The lid is lowered into the experimental chamber without touching the weights. This allows us to record the force needed to move the machine without deforming any experimental material. The raw force data from the background experiment is shown in Figure 5. All force data from the background experiment are considered to be machine noise.

Multiple machine noise signatures are identified in the raw force data from the background experiment. Low-frequency

oscillations in the raw data (Figure 5) are due to imperfect contact between the lid and the outer cylinder of the material chamber caused by a minuscule eccentricity of the cast acrylic cylinders (Bogatz, 2021). This results in orientations of the rotating cylinder where there is more friction between the lid and the outer cylinder resulting in a larger force required to rotate the experimental chamber. This imperfection leads to approximately one low-frequency wavelength per rotation. Another effect of non-constant friction between the lid and material chamber are very high frequency force oscillations lasting seconds

to minutes (Figure 5). The high frequency jumps are caused by the spring oscillating in response to increased friction between the lid and the outer cylinder. In this case, an increase in friction causes a sticking event resulting in loading of the spring until slip occurs, reoccurring at a high frequency. Further, increased friction also manifests as irregular and sharp jumps in force surrounded by a relatively smooth force signal (Figure 5). Friction between the lid and the outer cylinder increases through time and therefore causes more stick-slip motion and irregular force jumps with increased strain in the experiments. This limits

the number of rotations in an experiment to approximately 20.

Other noise associated with normal machine operation includes abrupt decreases in the pulling force due to repositioning of the hydraulic lid. The position of the hydraulic lid is set at the beginning of the experiment. During the experiments, the lid sinks under its own weight and corrects its position approximately every 80 seconds (Bogatz, 2021). While the lid motion is only a fraction of a millimetre, it leaves a signature in the force data. Other abrupt decreases in force take place due to slipping

between the lid and the outer cylinder. These decreases are generally greater than 5 N and often occur before the irregular jumps in force and smooth force signal outlined above. In addition to sharp jumps in force, elastic loading of the spring is included in the force signal every time the motor is stopped and starts to move again. The loading is shown as a drastic increase in force magnitude at the onset of the data collection (Figure 5). Lastly, we observe a low amplitude, high-frequency force oscillations that occur throughout the experiment. The amplitude of these oscillations is less than 0.5 N in the background

experiment (Bogatz, 2021). These oscillations are extremely regular, repeat throughout the experiment, and are originating from the stepper motor.

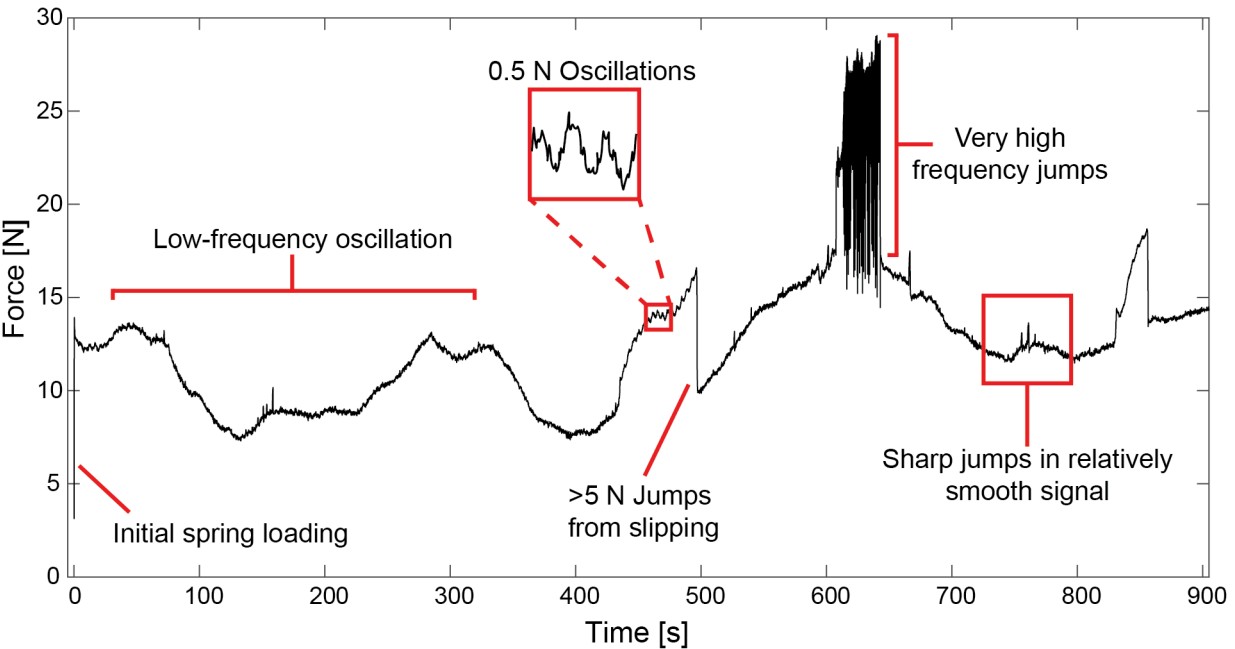

**Figure 5: Raw force data from the background experiment showing representative noise signals.**

## 4.3 Data processing

Identifying the different styles of machine noise above allows us to remove them from the bulk force recorded during an experiment. To process the pulling force data from an experiment, the raw force data is first separated into individual rotations (Figure 6 a). We then cut out large and recognizable noise events such as the noise associated with friction between the lid and the outer cylinder, as well as the abrupt decreases in pulling force described above and initial elastic loading of the spring. We are left with samples of the force data that only have the low-frequency oscillation and the regular low-amplitude, high-frequency oscillations, in addition to the force signal from the deforming materials (Figure 6 b). In a next step, we use a Lowess filter in Matlab to remove the regular low-amplitude, high-frequency oscillations as well as the force signal associated with material deformation from the sample (Figure 6 b). The Lowess filter is a non-parametric fitting tool that creates a linear regression for the data points contained within a specified window size (Bogatz, 2021). Using a window size of 100 data points to filter the data ensures only the low frequency wavelengths associated with the imperfect contact between the lid and the outer cylinder are preserved. We then take the difference between the filtered and the raw data of the samples (Figure 6 c).

At this point, the variance of the difference values can be calculated, illustrating the average spread of the data. Higher variance values are equivalent to larger differences between the total force measurement and the machine noise filter. In addition to the force signal from material deformation, the difference values also include the regular low-amplitude, high-frequency oscillations from the motor. However, this motor noise is present in all experiments and is removed from the calculated experiment variance values by subtracting the variance values of the background experiment (plotted in Figure 6 d). To visualize trends in the variance calculations, variance values from all samples in one rotation are averaged (e.g., Figure 6 d). A linear regression is fit though the points.

To calculate the average pulling force magnitude for each machine rotation we take the average of the data points in each rotation after removing the recognizable machine noise. The force magnitude for each rotation in the background force experiment is shown in Figure 6 e. The slight increase in force magnitude through time is due to an increase in friction between the lid and the outer cylinder with an increase in rotations. The background force magnitude trend is removed from the experimental results by subtracting the background force values from the force values obtained during experiments. Normalizing the experiment variance and force magnitude values for each rotation with the values calculated from the background experiment rotations ensures the remaining variance and force magnitude values are due to material deformation in the experimental apparatus.

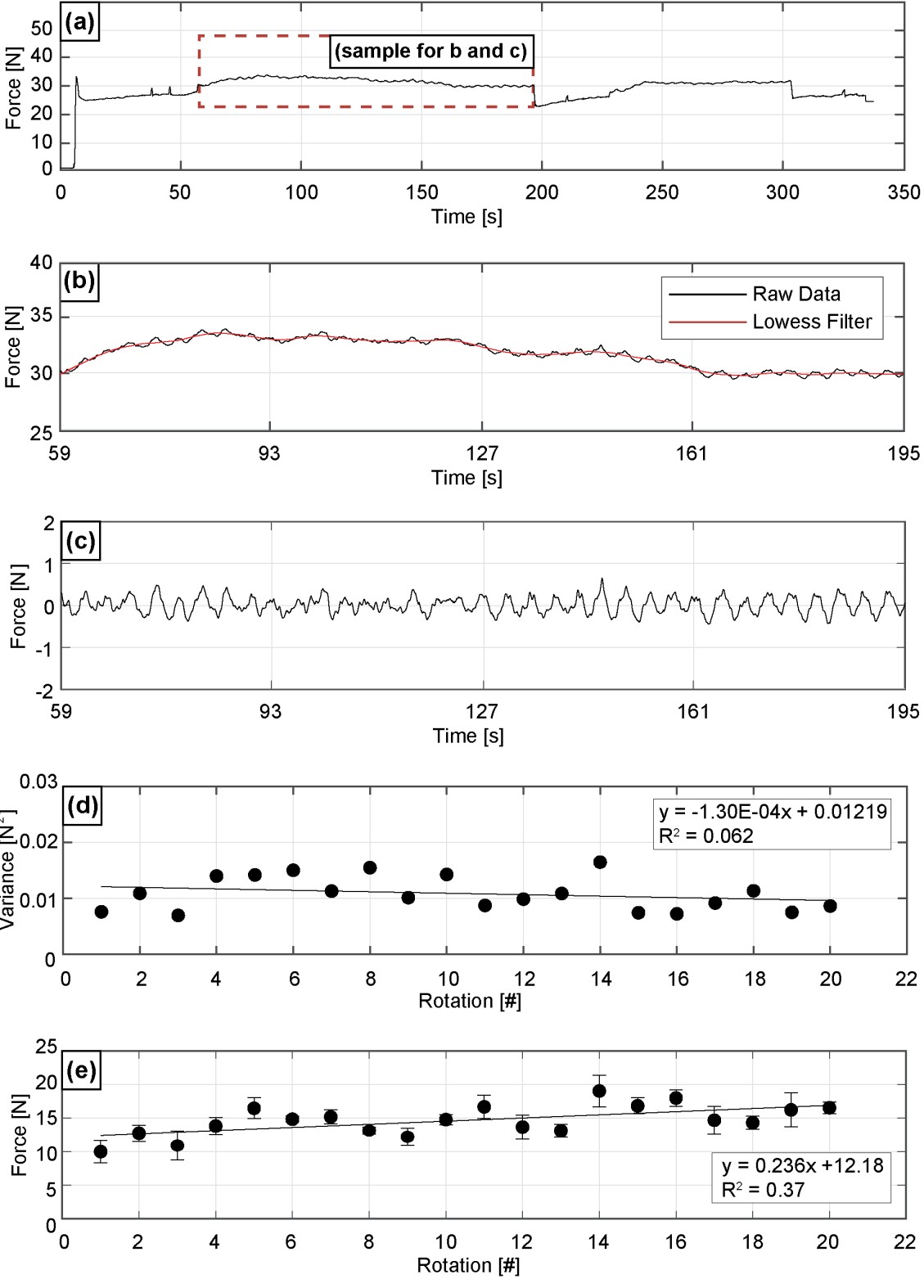

**Figure 6: Filtering process. a) Raw force data from one rotation of the semi-brittle experiment with 64 vol% HydroOrbs. b) Sample of force data without large and recognizable signal noise. Black line: Raw force data, red line Lowess filter. c) Difference between the raw data and filtered data. d) Change in variance over 20 rotations. Variance associated with regular machine operation is calculated from each rotation in the background experiment (no deforming experimental material present in the experimental chamber). e) Change in pulling force magnitude over 20 rotations associated with regular ring shear operation of the background experiment (no deforming experimental material).**

## 5 First experimental results

### 5.1 Results

We present results from four different experiments where we deform either a granular material (HydroOrbs), a semi-brittle material (mixture of HydroOrbs and Carbopol), or a viscous material (Carbopol). The small, punctured HydroOrbs made with DI water (Table 2) are used in the two brittle experiments and the semi-brittle experiment. The viscous and semi-brittle experiment include Carbopol with comparable yield stresses, viscosities, and shear thinning exponents. The aim of this section is not to present results from a comprehensive parameter study but rather to give an overview of what types of experiments have been conducted so far on this new shear machine and to stimulate discussion on future use and improvements.

An angular velocity of $0.019 \frac{rad}{sec}$ is used in all the experiments. Note, during an experiment the deformation strain rate can differ from the imposed rate of the machine due to the energy conserving boundary condition. The constant volume experiments are deformed by rotating the experimental chamber while the lid is held stationary. Figure 7 shows examples of the different types of experiments (brittle, semi-brittle, and viscous) before deformation (left column) and at the end of the experiment (right column).

We conduct two brittle experiments where the experimental chamber is filled with only HydroOrbs (Figure 8). Both experiments contain 2549 (Brittle 1) and 2706 (Brittle 2) of small, punctured DI orbs, respectively. The remaining pore space between the orbs is filled with air. Because air and the HydroOrbs have different light refraction indexes we cannot utilise the laser sheet to visualize the internal deformation and are limited to observations from the outside. When the lid is lowered between the two cylinders and onto the orbs to apply a normal force, the orbs deform elastically but do not fail. Note, the maximum confining pressure applied by the lid in either experiment is 0.80 kPa and 2kPa, respectively, and therefore less than the average yield stress of the HydroOrbs, which is 25.44 kPa. In the first brittle experiment (Brittle 1) individual orbs rearrange throughout most of the height of the experimental chamber with the start of deformation. Orb rearrangement continues throughout the 10 experiment rotations and no orb failure occurs. Only 10 rotations are completed in this experiment due to leaking of water from the orbs out of the chamber. The resultant raw force data records relatively large force oscillations when compared to the background experiment force data. These oscillations result in consistent variance values of approximately $0.1 \, N^2$ throughout the 10 rotations (Figure 8d). The force magnitude is also consistent throughout the rotations at around 10 to 12 N (Figure 8e).

The second brittle HydroOrb experiment (Brittle 2 in Figures 7 and 8) is conducted at a greater confining pressure of 2kPa. At the beginning of the experiment, all HydroOrbs are intact. They deform elastically when the normal force is applied but they

do not fail. Like experiment Brittle 1 at a lower confining pressure, with the initiation of shear, individual orbs rearrange throughout most of the height of the experimental chamber. However, unlike experiment Brittle 1, the orbs start to fail and break into smaller pieces near the shear boundary close to the stationary hydraulic lid (Figures 3 and 7). This breaking happens predominantly during the first six rotations. Orb fragments accumulate in a layer in the middle of the experimental chamber as the experiment progresses. Remaining orbs are counted after the experiment is completed to find 16.2% of orbs fractured and broke during the experiment. The force data of the brittle experiment records larger oscillations than the background experiment (Figure 8 a). The variance and the pulling force both decrease with an increase in number of rotations (Figure 8 d and e). The variance decreases from 0.2 $N^2$ to 0.1 $N^2$ over eighteen rotations. The pulling force decreases from approximately 30 N to 10 N. Only eighteen rotations are completed due to leakage of water from the experimental chamber.

The semi-brittle experiment contains 64 vol% HydroOrbs (2400 orbs) that are embedded in Carbopol. The Carbopol has a yield stress of 27.9 Pa and a viscosity of 236 Pa.s at a strain rate of 0.0207 $s^{-1}$. We follow the Carbopol preparation guidelines outlined in Birren and Reber (2019) and Di Giuseppe et al. (2015). To reduce the amount of bubbles entrapped in the Carbopol we directly mix the Carbopol in the experimental chamber. It is difficult to produce the Carbopol entirely bubble free. The shadow of these bubbles can be seen as dark horizontal lines in Figure 7 c and d. The confining pressure applied by the lid is approximately 1.5 kPa and is greater than the yield stress of the Carbopol but less than the average yield stress of the small DI water orbs used in the experiment (Table 2). Because the confining pressure is larger than the yield stress of Carbopol, Carbopol is deforming as a fluid. At the beginning of deformation, the majority of orbs are intact. Due to the mixing of the Carbopol, a few orbs can be damaged and show fractures. With increasing strain, some of the orbs start to fracture and break into smaller pieces. While some orbs break into two halves, others shatter (inset Figure 7d). At the end of the experiment approximately 6% of all orbs are broken in addition to orbs that broke during the experiment preparation. The number of fragments formed due to the breaking of orbs increased by 7% throughout the span of the experiment. Most of the fragment formation during the experiment occurred in the top third (vertical position within the illuminated cross-section) and outside third (horizontal position in the cross-section) of the experimental chamber where shear strain is greatest. Orb failure and fragment formation mostly took place from rotations 5 through 20. The filtered force curve is relatively smooth and does not significantly differ from the viscous experiment (Figure 8 b) except for higher pulling force values. The variance and pulling force do not change significantly between rotations (Figure 8 d and e). The variance is low at 0.04 $N^2$ while a pulling force of approximately 20 N is relatively high.

The yield stress and viscosity of the Carbopol at experimental strain rate are 26.6 Pa and 225.7 Pa.s, respectively in the viscous experiment. The yield stress of the Carbopol is again overcome by the confining pressure exerted by the hydraulic lid (1.4 kPa). Because the Carbopol is transparent we place mechanically passive markers in the gel to visualize deformation. Multiple strands of beads are added to the experiment at different locations between the two cylinders. One line of beads is located close to the outer wall of the experimental chamber, one in the middle and one close to the inner wall of the chamber. Because of the insignificant difference in density between the beads and the Carbopol they do not sink or rise. As expected, the lines of beads tilt with an increase in strain (Figure 7 e and f). We observe a shear band that accommodates the majority of the

deformation just slightly below the teeth of the lid. The strain recorded by the markers below the shear band is constant regardless of horizonal position within the experimental chamber. The recorded force data is rather smooth and does not show any prominent peaks (Figure 8 c). The variance shows no change with an increase in rotation (Figure 8 d). The pulling force required to deform the experiment is lowest of the three experiments at approximately 16 N and does not change with the amount of deformation (Figure 8 e).

## 5.2 Comparison and discussion of the experiment results

The experiment where we deform HydroOrbs only in the absence of fracturing (Brittle 1) can be used as the end member case of brittle/granular deformation. On the other hand, the experiment where we deform Carbopol only acts as the end-member example of purely viscous deformation. Both end-member cases show expected deformation patterns. In the brittle experiment (Brittle 1) the variance is relatively large as would be expected for a granular experiment where deformation is typically accommodated by rearranging of grains leading to stick-slip motion (e. g., Randolph-Flagg and Reber, 2020; Daniels and Hayman, 2009; Mair et al., 2002; Cain et al., 2001). As no orbs are failing, we do not observe any change in the variance values with increased rotations (e. g., Mair et al., 2002). The viscous experiment exhibits a rather smooth force signal. This is expected for a viscous material and corresponds to other observations of viscous flow (e. g., Reber et al., 2014). Comparing the other two experiments to the end-member cases allows us to determine how the change in experimental conditions or materials affects the deformation dynamics.

The brittle experiment at a confining pressure of 2kPa (experiment Brittle 2) is the only experiment that shows a significant decrease in variance and pulling force with an increase in number of rotations (Figure 8 d and e), which is the reason for using a non-linear trendline for the brittle data. Over multiple rotations, the force signal from the brittle experiment (Brittle 2) becomes smoother, which is illustrated in the decrease of the variance (Figure 8 d). We attribute this change to the decrease of breaking of orbs. Most orbs break during the first 5-6 rotations. The smaller orb pieces migrate towards the middle of the experimental chamber and form a band. A similar organization of grain fragments has previously been observed in high speed rotary experiments (Siman-Tov and Brodsky, 2018). During the first few rotations, the force magnitude recorded in Brittle 2 is larger than the values recorded in Brittle 1. This is due to the lower confining pressure used in the end member experiment (Figure 8e). Additionally, the force magnitude decreases drastically with increasing strain in Brittle 2 where orb failure occurs. In comparison, the force magnitude is relatively consistent in Brittle 1. The trendlines fit to the force magnitude data points of both brittle phase experiments meet by rotation 6 to 7. This coincides with most of the orb failure occurring in the first 6 rotations in experiment Brittle 2. The difference in the variance values recorded between the two brittle phase experiments also becomes smaller with an increase in rotations (Figure 8d). However, even with the power law trendline used for the variance values in Brittle 2, the variance trendlines between Brittle 1 and Brittle 2 never meet (Figure 8e), which may also be

due to a difference in confining pressure. The force data of Brittle 2 records the largest force oscillations of the four experiments resulting in a relatively large variance in the beginning rotations of the experiment.

The observations in the brittle experiments can be attributed to the rearranging and breaking of orbs during deformation. Both, rearrangement and breaking, can lead to a stick-slip like behaviour (e. g. Cain et al., 2001; Mair et al., 2002; Monzawa and

420 Otsuki, 2003). However, due to the 3D nature of the experiments, we record a force signal that integrates the effects of all material deformation taking place within the experimental chamber at any given time. We therefore do not observe sharp drops in force as would be expected of a typical stick-slip signal in a granular system (e. g., Randolph-Flagg and Reber, 2020; Daniels and Hayman, 2009; Mair et al., 2002; Cain et al., 2001) and we cannot resolve individual stick-slip events happening locally within the experimental chamber. Instead, we observe an overall noisier force signal due to bead rearrangement and breaking

throughout the experimental chamber, which is captured in the variance calculation (Figure 8a).

Across both brittle experiments, the variance values are greater than in either the semi-brittle or viscous experiments. In the viscous and semi-brittle experiments, the confining pressure applied by the hydraulic lid is larger than the yield stress of the Carbopol gel, leading to an entirely viscous response of the Carbopol to deformation. This matches the distributed deformation observed during the experiment. Distributed deformation in the Carbopol is in contrast to other studies where stick-slip signals

were recorded during Carbopol deformation (Birren and Reber, 2019; Reber et al., 2015). The absence of fracturing in the Carbopol is due to the experiment boundary condition (applied confining pressure is exceeding the yield stress in addition to the volume conserving boundary condition). Viscous deformation in the experimental chamber results in a bulk pulling force signal resembling creep with no large oscillations in force magnitude unlike what is observed in the brittle experiments. However, we do capture an increase in the variability of the force signal when compared to the background force experiment.

If the background force was equal to the force signal recorded during viscous experiments, the variance and force magnitude values plotted in Figure 8d and e would be zero. In all experiments plotted in Figure 8, the force magnitude and variance values are non-zero indicating material deformation is recorded by the force gauge.

Both the semi-brittle and viscous end member experiments show little change in the force magnitude and the variance with an increase in rotations. Despite the semi-brittle experiment containing closely packed orbs, their rearrangement and breaking

have no discernible impact on the force measurement. As soon as the pore space between the orbs is filled with a fluid (Carbopol) the resultant force measurements resemble the measurements of the viscous experiment (Figure 8 d and e) as opposed to the brittle end member experiment (Brittle 1) where orb rearrangement results in relatively large variance values. This is alike to observations of slip dynamics in lubricated granular experiments where small amounts of fluid smooth the stick-slip signal of the deforming granular material (Reber et al., 2014; Higashi and Sumita, 2009; Huang et al., 2005).

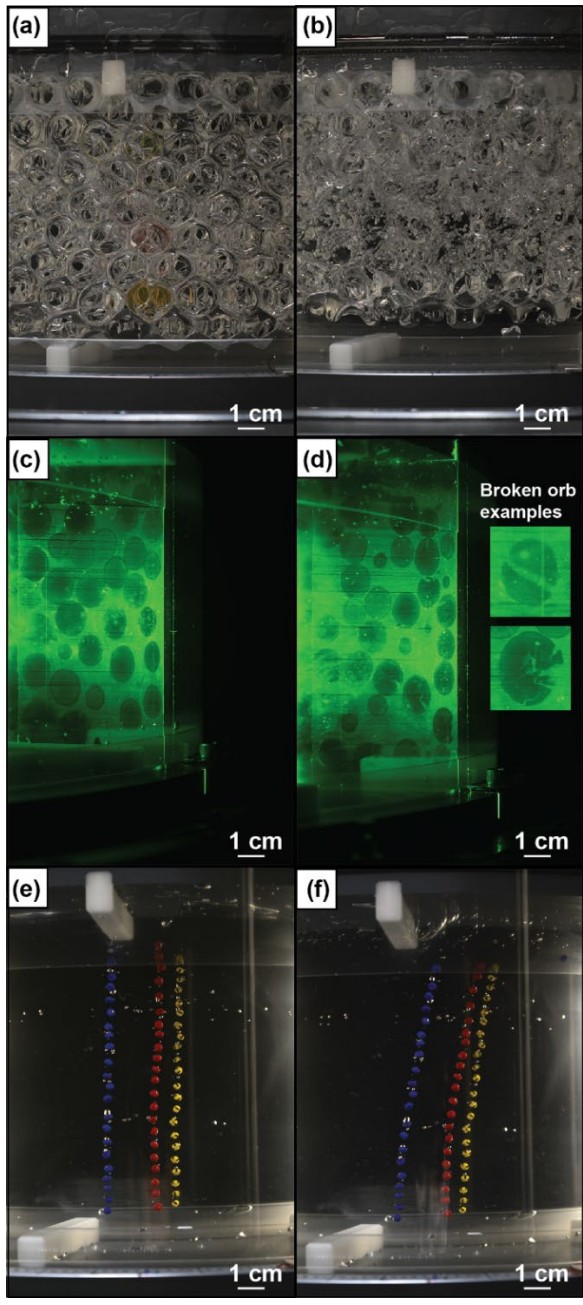

**Figure 7: Photographs of a brittle (Brittle 1) (a and b), semi-brittle (c and d), and viscous (e and f) experiment. Left column: at the beginning of the experiment before deformation, right column after 20 rotations.**

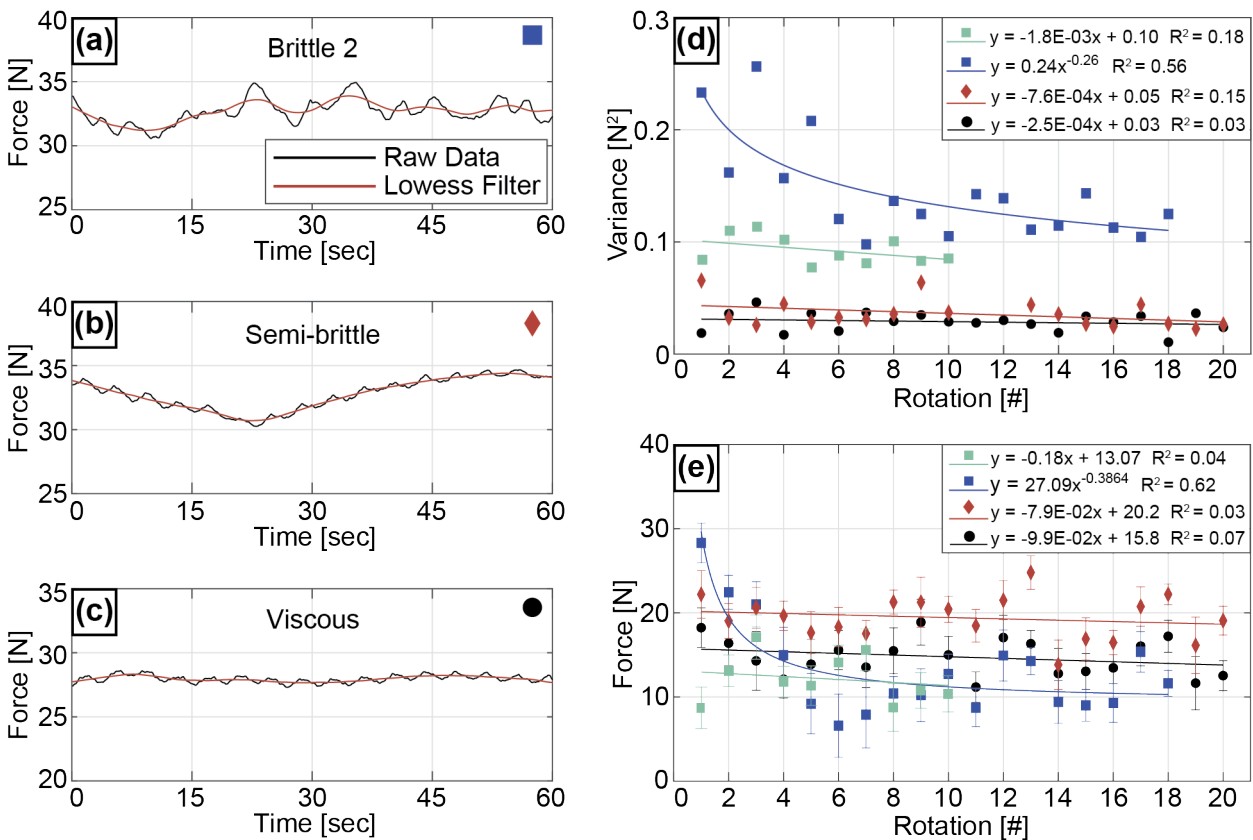

Figure 8: a) One-minute data sample of experiment deforming HydroOrbs (Brittle 2). b) One-minute data sample of semi-brittle experiment where 64 vol% of HydroOrbs are embedded in Carbopol. c) One-minute data sample of experiment deforming Carbopol (viscous). Black line is the raw data. Red line is the smoothed data. d) Change in variance with rotations for brittle, semi-brittle, and viscous experiments. e) Change in pulling force with rotations in brittle, semi-brittle, and viscous experiment. Teal squares in d and e represent experiment Brittle 1.

## 6 Limitations, potential improvements, and additions

As with every experimental approach there are multiple levels of limitations. We will focus here on the limitations associated with the experimental apparatus. While the experimental materials have their limitations too, especially when it comes to scaling of experimental findings to geological applications, they are strongly dependent on the exact research question.

A big drawback of the machine is that force can only be measured as a bulk property. This means that it is hard to filter out all the machine noise. This is also the reason why we cannot record a typical stick-slip signal from a deforming granular system. Furthermore, it remains impossible to record the force signal of a single breaking clast. The signal of an individual breaking clast would, however, be desirable especially for studies investigating the impact of failing brittle patches on slip dynamics. This could be addressed in the future by adding acoustic emission sensors into the experiment. Another limitation is that the

current design of the apparatus is unsuitable for experiments with a fluid that has a viscosity close to water. As the electrical motor is located at the same level as the bottom of the experimental chamber, leakage can become a problem that would potentially harm the motor. Future shear apparatus designs should consider mounting the motor at the level of the lid and applying the confining pressure from the bottom if experiments with low viscosity fluids are planned. Furthermore, low viscosity fluids can also have a limiting impact on the applied normal force as they are more prone to leaking. Another limitation associated with the motor is that its strong electromagnetic field can impact the force gauge measuring the normal force leading to increased noise.

Illuminated sections through the experiment can only be produced in experiments filled with light refraction index matched materials. As soon as there is a non-index matched phase present (most commonly air) this type of observation in not possible anymore. Furthermore, currently only one cross-section can be analysed at any given time, limiting the observation of the entire three-dimensional deformation.

## 7 Conclusion

We present the design of and first results from a new shear deformation apparatus for analogue multiphase experiments. The development of this experimental tool fills a gap in experimental capabilities to investigate multiphase deformation. The apparatus allows for recording of deformation dynamics ranging from stick-slip to creep. The experimental setup is designed for observations to be made of the internal deformation of an experiment in progress, giving insight into the three-dimensional nature of deformation. For cross-sectional observation through the experimental chamber, experimental materials that are light refraction index matched are used. We introduce three different experimental materials that fulfil this requirement. First experiments using these materials show the variability of force measurements and deformation styles. End-member experiments on a granular and a viscous material show the expected deformation dynamics. The presented experimental strategy has the potential to shed light on multi-phase processes associated with multiple geo-hazards.

## Data availability

All raw data can be provided by the corresponding author upon request.

## Author contribution

SML, HB, KB conducted the experiments, SML, HB, KB and JER analysed the results, and JER conceived the apparatus idea and was responsible for the designing and building as well as student supervision. All authors have contributed to the writing and figure drafting of this article.

## Competing interests

All authors declare that they have no conflict of interest.

## Acknowledgment

All authors were supported by NSF CAREER Grant #1843676 to JER. We thank Terry Hermann for engineering help and Neal Iverson for insightful discussions on custom deformation apparatus designs. We would like to thank Michael Rudolf and an anonymous reviewer for constructive questions and comments that increased the clarity of this contribution.

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
