# Peer review of "New ring shear deformation apparatus for three-dimensional multiphase experiments: First results"

_EGUsphere, 2022_

## Author Response (AR1)

**Responses to the Reviewer/referee comments**

All our responses are in blue.

**Michael Rudolf, referee 1**

**Summary**
The manuscript presents a novel ring shear tester with a transparent side wall to monitor deformation during shear. The machine is able to resolve the interaction of viscous and brittle phases by direct observation. Therefore, the machine has a high potential to improve the understanding of deformation in analogue experiments and can be used to improve the benchmarking of numerical models with experimental observations. The apparatus allows for several configurations that are desirable, such as constant force or constant rate boundary conditions as well as constant volume and constant normal stress experiments. This makes the approach very versatile and suitable to measure under many different conditions. An attachable spring allows for swift modification of the apparatus stiffness to modify the loading conditions such that stick-slip can be enforced or suppressed which is useful for measuring rate-and-state properties or similarly run stickslip experiments. The experimental setup and monitoring of the deformation is described clearly and would allow to rebuilt a similar machine in other laboratories. The qualitative description (section 5.1) of the deformation in the sample is good and clearly highlights the big advantage of this setup. However, I identify several problems with the presented experimental results which I outline as separate sections below.

**Calibration of Device**
The authors show results from experiments on HydroOrbs, HydroCubes, Carbopol, water and mixtures thereof. Table 2 shows the material properties but it is unclear whether these have been measured in this new device or in a separate apparatus. A newly developed measuring instrument should be calibrated on standardized materials so that the inaccuracies occurring with the measurement geometry can be clearly quantified. This could encompass a comparison of material properties from literature (e.g. Carbopol) with the parameters measured by the presented machine. Alternatively, other materials with well defined properties could be measured in this and another machine. Otherwise, a falsifying influence of the measurement geometry or design of the apparatus cannot be excluded. Therefore, I suggest that a quantitative comparison of at least one material is added to this manuscript to demonstrate that what the machine actually measures is valid. For example this could include a rheometric test of Carbopol or any other fluid (e.g. a silicone oil) without a spring at multiple loading rates (CSR-Test) to recover apparent viscosities.

The material properties presented in Table 2 were not measured in the ring shear apparatus. The presented new apparatus is not designed to measure material properties in the sense of a rheometer, instead its purpose is to investigate deformation dynamics and distributions.

We measure the rheology of the Carbopol gel with a rheometer, Rheosys Merlin VR. Samples are taken from every experiment to ensure consistency in both yield stress and viscosity values between experiments using the same viscous phase. Our measurements are comparable to measurements published in the literature (Di Giuseppe et al., 2015; Birren and Reber, 2019; Reber et al., 2015). The yield stress and elastic properties of the HydroOrbs and Cubes are measured using a force gauge and a caliper. Representative sample numbers (100 for each measurement) of orbs and cubes are used for each material property measurement. Again, our measurements fall within the range of published

values (Waitukaitis et al., 2017; Mukhopadhyay and Peixinho, 2011; Dijksman et al., 2017; Chang et al., 2018; James et al., 2020).

The purpose of experiments conducted with the new ring shear apparatus is to observe and quantify deformation of multi-phase materials. We are therefore not seeking to measure specific material properties (e.g., Carbopol viscosity or orb/cube yield stress) with the ring shear apparatus.
The here presented experimental setup, however, allows us to quantify and compare the bulk force needed to deform the experimental material. This allows us to investigate how changes in the experimental material is reflected in the force signal. Boundary conditions always impact measurements in any deformation setup. But by comparing different experimental materials and ratios we can make a statement on how these materials impact the deformation when experiencing the same boundary conditions. All the force gauges are calibrated and record the actual force.

Revision to the manuscript: We added details on how we determined the material properties of the experimental phases in the experimental materials section. We also added an explicit statement of the purpose of the new ring shear apparatus. Furthermore, in the discussion of the results we are now highlighting that we compare the experiments to the end-member cases of brittle/granular deformation and viscous deformation. Both end-member cases are behaving as expected and documented in the literature.

**Noisyness of Signal and Data**
The authors show the result of a background experiment with several possible sources of experimental noise and how each type of noise is treated. Fluctuations on a scale of 5 to 20 N are common over the whole experimental run. Additionally, there are several small scale noise components, such as the ones coming from the stepper motor. To me it is unclear what is signal and what is noise. For similar experiments done by the authors and coauthors the magnitude of stick-slip observations was in the order of 0.1 to 1 N. I would expect a similar range which would be undistinguishable from the extremly noisy signal coming from the machine. However, the measurement geometry was different for these experiments (Birren and Reber 2018), so what would be the expected magnitude of signal in the shear tester presented here? Additionally, how accurate is the load cell used here? From the manufacturers website I could see that most accuracy quantities (non-linearity, hysteresis, repeatability and offset) are in the range of .1 to 1 % of the rated output, which would fall into the region of the quantity that the authors would try to measure. Therefore, I am not sure whether the signal to noise ratio is actually high enough to discern between the signal (stick-slip) and external influences. This also leads to my last point:

The reviewer brings up a good point that led to significant clarifications in the manuscript. All force data recorded during the background experiment are considered noise. We identify and describe the different noise signals from the background experiment to outline how we cut-out, filter, and later subtract the background signals from the raw force data.

Previous experiments in Reber et al., (2015) and Birren and Reber (2019) investigate the stick-slip signal from deformation in the Carbopol gel in 2D shear experiments. In these experiments, the Carbopol deformed in a semi-brittle manner where the Carbopol in some cases fractures and in others flows. This results in a spectrum of deformation, where stick sip is associated with the formation of fractures in Carbopol and creep is associated with flow. The reviewer is correct that a stick-slip signal within the same range as reported by Reber et al, 2015 and Birren and Reber 2019 would be indistinguishable from the noise with the force acquisition and processing protocol of the ring shear device. The scope of the

two papers mentioned above, however, is significantly different. In the experiments presented here, we are using the Carbopol as the viscous phase. This means that we are not interested in the yield stress of the Carbopol but only in its subsequent flow. The confining pressures used in the ring shear device are high enough so that the Carbopol overcomes its yield stress right at the beginning of each experiment. We do not attempt to resolve stick-slip in the Carbopol itself stemming from its plastic rheologic component.
The reviewer makes a fair point concerning the force gauge accuracy. However, we do not attempt to resolve stick-slip in the Carbopol gel as the deformation in the gel is viscous in the ring shear apparatus.

Revisions to the manuscript: We added the comparison to these earlier studies using Carbopol to the discussion and we added the confining pressures for semi-brittle and viscous experiments to the experiment description.

**Processing of Experimental Results**
It is not clear to me what signal the authors are trying to extract from the data set. The filtering and noise processing applied is questionable. Am I correct that the final 'signal' is shown in Figure 6 c? To me this looks like pure noise, especially when considering the accuracy of the sensor. A better way to filter the signal in my view would be first to analyze the input time series with a frequency based method (FFT or continuous wavelet transformation) to see the magnitude and frequency of various noise components in the signal. Then a frequency based filter (High-Pass-Low-Pass etc...) could be designed to filter out specific frequencies from the signal improving the remaining signal. Contributions from the stepper motor and power grid frequency in the signal are then easily rooted out and removed from the signal. I understand from the manuscript that the variance is used as a simple proxy to how strongly the force fluctuates in the experiment and that the authors are able to recover a transient change in fluctuations over several rotations which could be related to the formation of a through going shear zone in the material. However, I am not sure whether this is an actual signal or just a processing artifact.

Figure 6c shows the difference between the raw data and the filtered data. This signal is then used to calculate the variance. We agree with the reviewer that the signal to noise ratio is low, making data extraction non-trivial. The suggestion of using FFT based filters is one we have investigated in the past. Beyond the irregularity in frequency of some of the larger machine noise signals, the frequency of the data from the deforming materials is very similar to the regular low-amplitude, high-frequency noise from the motor. Therefore, when a high/low pass filter is implemented to take out these noise frequencies, we also filter out a large part of the actual signal. Using averaged variance analysis allows us to analyze differences in the data despite an unfavorable signal to noise ratio.

By conducting background experiments (used for the filtering process) and multiple experiment repetitions, the observed differences in the data are not processing artifacts. These differences are not artefacts as we treat the data in all experiments exactly the same and they persist. Beyond that, we are observing differences in the force data that are systematic. The presented experimental results are representatives of a total of twenty experiments. There is a difference in force magnitude and variance values between the background experiment and experiments with deforming materials indicating that the material deformation in the machine results in a different signal. The difference between the background, the viscous, and semi-brittle experiments is small, but the difference between the background and brittle experiments is relatively larger. This leads to our interpretation that the viscous phase is able to dampen the signal stemming from frictional processes (as observed in the brittle

experiment). Furthermore, the force data is bolstered by the visual observations during the experiments that show the changes in deformation and breaking of orbs.

Revisions to the manuscript: We improved the explanations on data filtering. We added more detail to the discussion on what the force magnitude and variance signals mean in the context of the experiments.

**Concluding Remarks**
As such the manuscript provides a valuable asset to the analogue modelling community but several methodological improvements are required. Other than the above issues I have no major comments. A minor issue is that the style of the bibliography entries is not consistent and some are missing a DOI.

Fixed

The language is very good and I could not find any spelling or grammar issues. Therefore, I suggest major revisions because additional measurements are needed and a different data processing should be applied.

We would like to thank Michael Rudolph for constructive remarks and suggestions that have improved the manuscript.
* * *
In response to Anonymous Reviewer 2:

**Anonymous Reviewer**

**GENERAL COMMENTS**

McLafferty and coauthors present a detailed description of a novel shear deformation apparatus capable of deforming multiphase materials. The apparatus features a transparent chamber allowing visual inspection of the sample during the measurements. The first part of the article report information about the experimental apparatus, boundary conditions and materials tested. The second part is about calibration, measurement of different materials and potentialities/limitations. The manuscript is well structured and clearly written. Figures are very clear. This type of apparatus is not representing a novelty for the community of modellers of tectonic processes. However, previous instruments focused on granular materials characterization, while the apparatus (in particular the lid characteristics) of McLafferty et al., allows testing also viscous materials. To my knowledge, this represent a novelty at the boundary between rheometry and granular materials characterization. Despite this type of papers generally appear very technical, I found them very useful for understanding the underlying physics. As this represents and important step for the community, I recommend being extra careful (I appreciate the details for this initial submission). However there are a couple of points that deserve attention: calibration and noise (see specific points). I'm confident the Authors can address these issues as they might have run complementary measurements that can help making more robust this study. I report few specific comments below (ranked in order of importance) and minor importance recommendations as to how to improve the manuscript. Hope my comments help improving the ms.

**SPECIFIC COMMENTS**

**Start with a standard:** Authors can use a standard material like silicone oil, and use it to calibrate their shear tester. I understood this machine might have problems with low viscosity fluids (like water), but what about glucose syrup for example? This material has linear-viscous strain independent and shear rate independent rheology with viscosity in the order of few hundreds Pas, similar to the Carbopol, and should be quite easy to measure. Authors should compare the results obtained with the shear tester e.g., with the study by Schellart (2011). If the problem is finding a material with yield stress then they should compare their results (now reported in terms of Force e.g., in figure 8) with the study about Carbopol rheological characterization by Di Giuseppe et al (2015). I think this step is very important and helpful for a suspicious reader. If testing a viscous material is problematic, then try with a granular one with well known properties and compare them with your instrument output.

The purpose of this new apparatus is to observe and quantify deformation of mixtures of experimental materials with known physical properties. This method allows us to compare the impact of the experimental material properties and relative amount of material phases on deformation dynamics and distribution. We therefore are not seeking to measure specific material properties (e.g., viscosity of Carbopol or yield stresses of the brittle phase) with the ring shear apparatus itself. We instead measure the properties by other means (rheometer) before placing them into the ring shear apparatus.

We use materials with known properties and compare our own measurements to these studies to ensure consistency. The yield stress, viscosity, and shear thinning exponent of the Carbopol gel used in this manuscript are measured using a rheometer (Rheosys Merlin VR). The values presented in this study are comparable to those found in Di Giuseppe et al., (2015), Birren and Reber (2019), and Reber et al., (2015). In terms of the brittle phase, the yield stress and elastic properties of the HydroOrbs and HydroCubes are determined using a force gauge and caliper. The non-punctured yield stress values of the orbs found in this study fall within the range of 75 to 175 kPa measured under variable loading rates by James et al., (2020). Additionally, the elastic properties of the orbs are consistent with previous studies, where the Young's moduli range from 10 to 100 kPa (e.g., Waitukaitis et al., 2017; Mukhopadhyay and Peixinho, 2011; Dijksman et al., 2017; Chang et al., 2018; James et al., 2020), and Poisson's ratio range from 0.3 to 0.45 (Chang et al., 2018). All material property measurements are conducted on a representative sample size of orbs and cubes.

Experiments involving only the viscous phase (Carbopol) or only the brittle phase (HydroOrbs) serve as our end member experiments. In these end member experiments, we know the properties of the Carbopol and orbs and can use the visual observations and force data values as a benchmark to compare with multiphase experiment results. In the case of the viscous experiment, shear of the Carbopol gel results in distributed strain below the shear band. As expected for a viscous fluid, no significant oscillations are observed in the force data. These two observations are expected for any viscous fluid deformed in the ring shear apparatus. We tested multiple different viscosities and obtained the same results and therefore decided to not include them in the manuscript but only show a representative experiment.

The comments from both reviewers have motivated us to include results from a granular experiment where HydroOrbs are deformed without breaking. This experiment now acts as the benchmark for the brittle material phase. If no grain failure is present, the force and variance values are consistent in magnitude throughout the experiment as expected. Presenting both brittle experiments now allows us to show how the variance and force magnitude change depending on grain failure.

Quantitatively comparing our measurements to measurements obtained by others is non-trivial as other studies measure different quantities (resistance to shear, torque, etc.), while we measure pulling force with our own machine noise. We do however discuss our results qualitatively in light of previous research.

We are presenting in this contribution the benchmark experiments obtained with the ring shear apparatus (the Carbopol only experiment, and the brittle experiment without failure). This allows us to compare how material ratios etc., impact the measurements. If, in the future we are interested in changing the experimental material, we will of course have to run similar benchmark experiments again. Testing other materials at this point is outside the scope of this contribution.

Revisions to the manuscript: A statement on the purpose of the new ring shear apparatus at the beginning of the apparatus design section has been included. Additional information on material properties, how we measure them before experiments, and how they compare to other studies are added. We also add an additional brittle phase experiment where no orb failure occurs to serve as the brittle end member experiment. We modified the discussion and comparison section to reflect these changes.

**Noise:** Is it possible to add information about load cell characteristics? A reader should be helped in understanding what is noise and what is signal. The reader is confused. E.g., at line 311 we understand there is no significant difference between measurement of a given material and "background" signal. If semibrittle fig8b is not different from background (I understand fig 5), than what is a reliable signal from deforming material?

The reviewer brings up a good point that we have used to revise the manuscript. The machine noise is anything measured in the background experiment (Figure 5). We use the background experiment to identify the machine noise that we later manually cut out and then filter with the Lowess filter. The background experiment goes through the same filtering process as the experiments with deforming materials. This allows us to subtract force magnitude and variance values calculated from the background experiment from those calculated from the experiments with deforming materials (Figure 6d and e). The averaged force magnitude and variance values from each rotation of an experiment after the respective background values are subtracted are shown in Figure 8d and e. Figure 8a, b, and c show the raw, unprocessed data from the experiments for qualitative comparison, while Figure 8d and e show experimental data without background noise for reliable quantitative comparison.

Line 311 has a mistake, thank you for catching it. We are not comparing the semi-brittle experiment to the background experiment but to the viscous experiment. We fixed this mistake.

Revisions to the manuscript: We revised the background experiment, machine noise, and force data processing sections of the manuscript significantly and hopefully improved its clarity. We have also revised line 311.

**3D observation:** Authors report a couple of times they see 3D deformation with a light sheet, but they report only 2D pictures of the deforming material with this cross section. Tracking 3D deformation could be feasible but very difficult to implement in this setup (maybe with stereo PIV?!). At this stage Authors report only 2D qualitative visual information. See line 369… true only one cross section.

We use the laser sheet to observe the third dimension in the experiment. We include photos of the 2D cross section in Figure 7. The photos in Figure 7 are representative of deformation occurring throughout the ring shear apparatus. We illuminate a cross-section with a laser sheet that is stationary and move the experimental chamber through the light sheet during and after every rotation. As we move the experimental chamber through the stationary laser sheet, we take photos of the illuminated cross-section at every one degree around the chamber for a total of 360 photos. Using these photos, we count the number of broken orbs and fragments and record their position in the experimental chamber. By recording orb deformation in the cross-section photos, we are able to track deformation in the brittle phase around the experimental chamber with increasing rotations.

However, there are limits to 3D observations of deformation with this laser method. Line 369 (in the limitation section of the manuscript) states there is only one cross-section position (the laser sheet stays stationary). This limits the real-time 3$^{rd}$ dimension observations of deforming materials in other sections of the ring shear apparatus.

Revisions to the manuscript: We revised the description of how we observe the 3$^{rd}$ dimension and how data are recorded from the 360 photos around the experimental chamber (visual documentation section). We added how we obtain and treat quantitative results of broken orb and fragment percentages as well as broken orb locations in the semi-brittle experiment (results section).

**About HydroCubes:** re-consider the added value of describing this material. None of the samples tested and reported in paragraph 5.1 include this material. It would be enough just mention their existence without entering in the details. Especially if this material is not used in the current version of the ms.

Referee 2 makes a good point. None of the experiments presented here involve the HydroCubes. However, HydroCubes are an example of a material that can be used in the apparatus that is indexed matched for laser illumination.

Revisions to the manuscript: We decreased the level of detail on HydroCubes in the text. Instead we simply state HydroCubes are another option for a brittle phase material and list properties in the table.

**TECHNICAL CORRECTIONS**

**line 128**: "almost identical indices" sounds not really professional. Can Authors specify how much similar?

Revision to manuscript: We added the refractive indices of Carbopol, HydroOrbs, and acrylic walls of the cylinders to the manuscript.

**Line 170:** small and large sizes. Please specify the size. This information could also be added in table 2. At the moment the reader can oly understand they are roughly 1 – 2/3 cm from figure 4. Can the Authors provide precise measurements?

Revisions to the manuscript: Diameters of the HydroOrbs resulting from the small and large dehydrated plastic pellets are now listed in Table 2.

**Line 175**: specify what DI means. Deionized?

Revision to manuscript: We added (DI) after the first use of the word deionized.

**Table2:** check density of carbopol (reported in kg/m3?)

Revision to manuscript: Changed Carbopol density to match the correct units listed in the left-most column (1.01 to 1.03 $\frac{g}{cm^3}$).

**Table2:** about viscosity of carbopol. Is it possible to add information about this range? Is it associated to strain rate variation? Or pH…

This range is associated with the viscosity values of the Carbopol gel used in experiments to date. The range listed is achieved by changing the concentration of the polymer while keeping the pH constant.

**Line 258:** window size of 100 what? Seconds measurement points?

Revision to the manuscript: Added the window size corresponds to 100 data points.

**Line 285:** specify if HydroOrbs reported in paragraph 5.1 are punctured (lower yield stress) or not.

Revisions to the manuscript: The HydroOrbs used in the brittle experiments and semi-brittle experiment are the small, punctured HydroOrbs made with DI water (added to the first paragraph to the results section).

**Line 243:** check the information about 20 rotations in the text but only 18 are shown for brittle material in figure 8.

Variance values from the brittle experiment shown in Figure 8 (now labeled as Brittle 2) are calculated for 18 rotations. 20 rotations are the standard procedure for all experiments, but brittle experiments sometimes result in water leakage cutting the experiment short. We have to stop the experiment when water starts to leak out of the experimental to ensure the motor does not get damaged.

Revision to the manuscript: Outlined why only 18 rotations were completed instead of the usual 20 for experiment Brittle 2.

**Line 306:** impossible or difficult?

Revision: Difficult

**Line 328:** we do not see stick-slip behavior. Explain why. Is it because of the instrument stiffness (maybe lower the spring then), or because your measurement integrate data from the whole "ring". I think it could be useful to show that in the revised ms.

We do not observe sharp drops in force recorded in the raw force data as would be expected of a typical stick-slip signal in a granular system (e.g., Randolph-Flag and Reber, 2019: Mair et al., 2002; Cain et al., 2001; Daniels and Hayman, 2008). This is due to the 3D nature of the experiments and the integration of all force signals from material deformation around the experimental chamber.

Revision to the manuscript: We added further discussion on the force signal from the brittle experiments to the discussion section of the manuscript and clarified this point.

**Line 359:** are the Author sure about detecting acoustic emissions from a gelly-like material!? In this case photoelasticity can be useful…

Yes, work on acoustic emissions in HydroOrbs is forthcoming :)

We thank Referee 2 for their comments and suggestions that helped improve this manuscript.